# Parent Support Is Related to Physical Activity among Children and Youth with Disabilities during the COVID-19 Pandemic: Findings from the National Physical Activity Measurement (NPAM) Study

Maeghan E. James [1], Nikoleta Odorico [1], Sarah A. Moore [2], Kathleen A. Martin Ginis [3,4,5], Rebecca L. Bassett-Gunter [6] and Kelly P. Arbour-Nicitopoulos [1,*]

1 Faculty of Kinesiology and Physical Education, University of Toronto, Toronto, ON M5S 2W6, Canada; maeg.james@mail.utoronto.ca (M.E.J.); niki.odorico@mail.utoronto.ca (N.O.)
2 School of Health and Human Performance, Dalhousie University, Halifax, NS B3H 4R2, Canada; sarah.moore@dal.ca
3 Department of Medicine, Faculty of Medicine, University of British Columbia, Kelowna, BC V5Z 1M9, Canada; kathleen_martin.ginis@ubc.ca
4 School of Health and Exercise Sciences, University of British Columbia, Kelowna, BC V1V 1V7, Canada
5 International Collaboration on Repair Discoveries (ICORD), University of British Columbia, Vancouver, BC V5Z 1M9, Canada
6 Faculty of Health, School of Kinesiology & Health Science, York University, Toronto, ON M3J 1P3, Canada; rgunter@yorku.ca
* Correspondence: kelly.arbour@utoronto.ca

**Abstract:** Physical activity (PA) among children and youth with disabilities (CYD) has been negatively impacted by the COVID-19 pandemic. Parent PA support and parent PA modelling (i.e., parents engaging in PA themselves) have been shown to be associated with PA in CYD. However, parents' influence on the PA behaviours of CYD during the pandemic remains unknown. The purpose of this study was to examine the relationship between parent PA support and parent PA modelling (i.e., parent moderate-to-vigorous PA (MVPA)) and the PA behaviours of CYD. It was hypothesized that higher levels of parent PA support and parent PA modelling would significantly relate to both child MVPA and child PA at any intensity. An online survey was sent to parents of CYD in November 2020 (i.e., during the second wave of the COVID-19 pandemic in Canada) that assessed the MVPA and total PA (any intensity), parent PA support (e.g., encouraging PA, providing transportation for PA), and parent MVPA. Separate linear regression models assessed the relationships between parent PA support and parent PA modelling with (a) child MVPA and (b) child PA at any intensity. Parent and child age, child gender and disability group, marital status, and household type were controlled for in all analyses. A total of 86 parents (Mage = 43 years, SD = 5.9; 93% mothers) of CYD (Mage = 11 years, SD = 3.3; 20% girls; 77% with a developmental disability) completed the survey. Parent PA support was significantly associated with child MVPA (β = 0.30, CI = 0.067–0.438, *p* = 0.008) but not child PA at any intensity. No significant relationship was shown between parent PA modelling and either child MVPA or child PA at any intensity. Findings suggest that parent PA support, but not parent PA modelling, was associated with PA in CYD, at least during the acute period of the pandemic. Greater efforts must be directed towards developing effective parent PA support interventions to reduce the detrimental effects of the COVID-19 pandemic on PA in CYD.

**Keywords:** COVID-19; children and youth; disabilities; physical activity; parent support

## 1. Introduction

Physical activity (PA) is an essential component of child development that provides many physical [1], cognitive [2], and psychosocial [3] health benefits. Children and youth

with disabilities (CYD) often exhibit poorer physical and mental health compared to children without disabilities [4–6] and engage in less PA [7]. Engaging regularly in PA, while important for all children and youth, may be particularly important for health outcomes among CYD. For children and youth without disabilities, strong evidence exists for the engagement in at least 60 min of moderate-to-vigorous PA (MVPA) per day, as it relates to optimal health benefits [8]. While most CYD are not meeting the recommendation of at least 60 min of MVPA per day [7], research shows that CYD spend more time engaging in light-intensity PA which is also important for health [9]. Therefore, efforts should be made to promote all intensities of PA among CYD in addition to MVPA.

The COVID-19 pandemic introduced several public health restrictions including facility closures (e.g., schools, recreation centers, physiotherapy clinics), limiting the size of group gatherings (e.g., children's sport programs), and work-from-home mandates that made it difficult for families to uphold regular daily routines. Unfortunately, these restrictions were associated with decreases in PA levels in typically developing children and youth [10,11]. Restrictions may be that much more detrimental to CYD given that COVID-19 disproportionately affected people living with disabilities (e.g., reduced service provision for CYD, lack of inclusive health resources) [12]. Importantly, CYD have been shown to be particularly vulnerable to the impact of COVID-19-related restrictions as it pertains to PA [13,14]. For example, a recent national survey of CYD in Canada conducted by Moore et al. [14] demonstrated that only 5.3% of CYD were engaging in the recommended levels of MVPA for health benefits during the first wave of the COVID-19 pandemic.

While there are several potential contributing factors to children's PA participation (e.g., the child's previous engagement in and enjoyment of PA) during the pandemic, parent PA support may be one factor that plays a role. Parent PA support has been defined as the level of facilitation parents provide to their child to engage in PA, including both tangible (e.g., transportation, financial support) and intangible (e.g., information, encouragement) forms of support [15]. Parent PA support has consistently shown a positive relationship with PA in CYD [16–20]. Research conducted prior to the COVID-19 pandemic suggests parents (i.e., the primary caregiver of the child who may or may not be the biological parent) play a critical and, in many cases, laboured role in supporting the PA of CYD [16,21–23]. Parents of CYD are often responsible for seeking out suitable/inclusive PA opportunities for their children, resulting in both physical and emotional labour [22,24]. Additionally, when PA opportunities are available, responsibility is typically placed on parents to support their child's participation and ensure that their child is provided with the same quality experience as children without disabilities [22]. Indeed, parent PA support is critical to facilitating PA among CYD.

One unique form of support that is of particular importance is parent PA modelling (e.g., parents engaging in PA behaviours themselves). Research in typically developing children demonstrates that when parents model PA, their children are more likely to meet PA recommendations (i.e., at least 60 min of MVPA per day) [18,25]. For example, Yazdani et al. [26] found that among CYD (primarily neurodevelopmental disabilities) between the ages of 4 and 21 years, those whose parents engaged in at least three hours of PA per week were more likely to be physically active. Taken together, these studies highlight parent PA support and PA modelling as important factors to consider for facilitating PA in CYD and therefore a potential PA correlate for CYD PA engagement during the COVID-19 pandemic.

Opportunities to be physically active in the community during the pandemic have been impacted for children and youth and, at times, resulted in greater reliance on their parents to support their PA (e.g., virtual programming or parent-organized PA). Conceivably, the way parents of CYD are supporting and modelling PA during the COVID-19 pandemic may differ compared with pre-pandemic support and modelling. A national study of parents of children and youth living in Canada indicated that parent PA support and parent PA modelling had a weak-to-moderate positive association with children's PA participation during the COVID-19 pandemic [10,11]. A limitation of this research is its lack of focus on the relationship between parent PA support and parent PA modelling on child PA in

CYD. This is an important gap to address given that parent PA support for CYD is often more labour-intensive compared to parents of children without disabilities [22]. Recently, with the COVID-19 pandemic and associated public health restrictions, this labour is even more cumbersome due to the virtual delivery of PA programs and lack of one-to-one support. For example, for some CYD to engage in virtual programming, parents may need to physically support their child at home as they participate in PA [13]. Moreover, parents may need to emotionally encourage their child's participation, as virtual experiences have been found to be less enjoyable for CYD when compared to in-person programming [27]. While parents of CYD have indicated they have been more involved with their child's PA during the pandemic, parents have also expressed difficulties in their ability to facilitate PA opportunities that meet the unique needs, interests, and abilities of their child [13]. Esentürk [27] demonstrated this difficulty whereby parents of CYD indicated the need for more education, expert support, and resources specific to CYD to optimally support their child's PA.

The COVID-19 pandemic and associated public health restrictions have resulted in several PA barriers for CYD and new responsibilities for parents [28]. The World Health Organization has explicitly recommended for parents to be actively involved in their child's movement opportunities at home to combat the toll of the pandemic on the health and well-being of young children [29,30]. Yet, the influence of parent PA support behaviours on the PA of CYD during the COVID-19 pandemic has yet to be examined. This is a critical research gap that needs to be addressed to identify potential strategies to increase the PA participation of CYD throughout pandemic recovery and as these restrictions loosen. The present study aimed to address this gap and contribute to the broader parent PA support literature by examining the relationship between parent PA support behaviours during the COVID-19 pandemic and the PA behaviour of CYD. In line with previous research on parent PA support [16–18,27], it was hypothesized that both parent PA support and parent PA modelling would significantly predict the PA behaviour of CYD during the COVID-19 pandemic.

## 2. Methods

### 2.1. Study Sample and Data Collection

Study participants were part of a larger cross-sectional investigation called the National Physical Activity Measurement (NPAM) study [30]. The NPAM study aims to examine the movement behaviours of school-aged CYD in Canada. Study eligibility criteria for the NPAM study include: (a) being a parent of school-aged (4 to 17 years) CYD (b) living in Canada and (c) proficient in English or French. In April 2020, parents who completed the NPAM study were invited to take part in a secondary COVID-19-related study. This secondary study consisted of the online administration of the 2020 COVID-19 and Childhood Movement Behaviours Survey (developed by Moore et al. [10]) which included questions regarding family demographics, current and previous movement behaviours, and the impact of COVID-19-related restrictions on the movement behaviours and health of families. Parents were invited to complete the survey at two time points, May 2020 and November 2020, using REDCap® to assess the acute [14] and longer-term associations [13] of the pandemic on the movement behaviours of CYD in Canada. The current study examined the data collected from the subset of parents ($N$ = 86) who responded to the November 2020 COVID-19 survey invitation and for which parent PA support and parent PA data were available. Institutional research ethics approval was obtained from the University of Toronto prior to any data collection, and parents provided informed consent at the beginning of the survey.

### 2.2. Survey Content

The survey consisted of an amended version of the 2020 Childhood Movement Behaviours Survey [10]. Demographic information collected included: parent and child age, gender and ethnicity, child's disability group, parent employment status, household

income, household type, and marital status. Parents indicated their child's disability group as: (1) developmental, (2) physical and/or sensory, or (3) other. Household type was dichotomized into detached home or other dwelling type (e.g., apartment), and child age was dichotomized as either child (4–11 years) or youth (12–17 years). Parent PA support behaviours were measured using three items. Each item asked parents to rate their level of support on a 5-point Likert scale ranging from "a lot less" to "a lot more". Parents were asked to indicate how often, compared to before the onset of the COVID-19 pandemic, they: (a) encouraged their child to participate in PA or sport, (b) played outside with their child or did PA or sport with their child, and (c) drove or provided transportation for their child to do PA or sport. The scores for each item were totalled to produce an overall parent PA support score ranging from 0 to 15. To assess child PA behaviours, parents were asked to indicate over the past week, how many days their child engaged in at least 60 min of PA at any intensity. Parents were also asked specifically about days their child engaged in at least 60 min of moderate-to-vigorous physical activity (MVPA). These two PA items were treated as continuous variables for the analyses. One item was used to assess parent PA modelling (i.e., PA levels among parents). This item asked parents to indicate the total time (in minutes), over the last week, they engaged in MVPA, which aligns with the intensity for which adults are recommended to engage in for health benefits [31].

*2.3. Statistical Analysis*

Descriptive statistics were calculated for each variable. Continuous variables were expressed as mean (standard deviation) and categorical variables as frequency (percent). Bivariate Pearson correlation analyses were used to examine the relationships between parent PA support, parent PA modelling (total minutes of MVPA), child PA (both the number of days children and youth spent in at least 60 min of MVPA and PA at any level) and child age, child gender, and household type. Bivariate Spearman's rank correlation analyses were used to examine the relationship between disability group and parent PA support, parent PA modelling, and child PA (both the number of days children and youth spent in at least 60 min of MVPA and PA at any level). Two separate multiple linear regression models were conducted to examine parent PA support and parent PA modelling as predictors of (1) child PA at any level and (2) child MVPA. Child age, child gender, child disability group, and household type were included as covariates in both regression models. Model 1 regressed child PA at any intensity onto parent PA support, parent PA modelling, and the four covariates. Model 2 regressed child MVPA onto parent PA support, parent PA modelling, and the four covariates. A forced-entry method was used for both linear regression models. A probability plot (P-P) was used to test for the assumption of normality. Collinearity was assessed using variance inflation factors (VIFs) of the independent variables; VIFs of 10 or more suggested collinearity. All statistical analyses were conducted in SPSS version 26.0 (IBM, Armonk, NY, USA), and significance was set at a two-tailed alpha value of $p < 0.05$.

## 3. Results

*3.1. Sample Demographics*

Participant characteristics are presented in Table 1. In total, 86 parents completed the survey in November 2020. The majority of the sample included mothers (91%, $M_{age}$ = 43 years, SD = 5.92) who were married (81%) and were living in detached homes (78%). With regard to the CYD, 20% of the sample were girls and, on average, 11 years old (SD = 3.26). Approximately 77% of the children and youth were reported by their parents to have a developmental disability, and the remaining children and youth were reported to have a physical or sensory disability (14%) or "other" (9%).

**Table 1.** Parent and child demographic characteristics (*N* = 86).

| Parent Demographic Profile | |
|---|---|
| Age (years), *M* (SD) | 42.42 (5.74) |
| Gender, Women, *N* (%) | 131 (92.1) |
| Ethnicity, *N* (%) | |
| Caucasian | 65 (75.6) |
| East Asian (e.g., Chinese, Japanese, Korean, Taiwanese) | 7 (8.1) |
| First Nations | 1 (1.2) |
| Hispanic (e.g., Central American, South American, Mexican, Spanish) | 1 (1.2) |
| South Asian (e.g., Indian, Pakistani, Sri Lankan) | 3 (3.5) |
| Southeast Asian (e.g., Cambodian, Indonesian, Laotian, Vietnamese) | 3 (3.5) |
| Western Asian (e.g., Afghani, Armenian, Egyptian, Iranian, Iraqi, Saudi Arabian, Turkish) | 1 (1.2) |
| Mixed ethnicity | 2 (2.3) |
| Prefer not to answer | 3 (3.5) |
| Marital Status, *N* (%) | |
| Single | 6 (7.0) |
| Married | 72 (83.7) |
| Separated | 3 (3.5) |
| Divorced | 3 (3.5) |
| Undisclosed | 2 (2.3) |
| Annual Household Income, *N* (%) | |
| <$50,000 | 13 (15.1) |
| $50,000 to $99,000 | 21 (24.4) |
| >$100,000 | 38 (44.2) |
| Undisclosed | 14 (16.3) |
| Employment Status, *N* (%) | |
| Full-time | 37 (43.0) |
| Part-time | 20 (23.3) |
| Self-employed | 9 (10.5) |
| Unemployed | 16 (18.6) |
| Other (e.g., student, retired) | 3 (3.5) |
| Undisclosed | 1 (1.2) |
| Household Type, *N* (%) | |
| Apartment or condominium | 5 (5.8) |
| Detached home | 68 (79.1) |
| Semi-attached home | 4 (4.7) |
| Townhouse | 9 (10.5) |
| Parent Support for Physical Activity, *M (SD)* | |
| I have encouraged my child to participate in physical activity or sport | 3.4 (1.1) |
| I play outside with my child or did physical activity or sport with my child | 3.2 (1.0) |
| I drove or provided transportation for my child to do physical activity or sport | 2.7 (1.2) |
| Total parent support for physical activity | 9.3 (2.5) |
| Child Demographic Profile | |
| Age (years), *M* (SD) | 11.13 (3.26) |
| Children (aged 4 to 11 years) (*n* = 46) | 8.57 (1.83) |
| Youth (aged 12 to 17 years) (*n* = 40) | 14.08 (1.61) |
| Gender, Girl, *N* (%) | 17 (19.8) |
| Disability Group, *N* (%) | |
| Developmental | 66 (76.7) |
| Physical and sensory | 12 (14.0) |
| Combination | 8 (9.3) |

### 3.2. Associations between Parent PA Support, Parent PA Modelling, Child PA and Family Demographics

Bivariate correlations between parent PA support, parent PA modelling, child PA (at any intensity and MVPA specifically), and the four demographic correlates are shown in Table 2. Parent PA support and parent PA modelling were not significantly correlated ($r = 0.20$, $p = 0.072$). For child MVPA, parent PA support was significant and positively correlated ($r = 0.28$, $p = 0.012$). No significant correlation was shown between parent PA modelling and child MVPA ($r = -0.02$, $p = 0.831$). No demographic variables were found to be significantly correlated with child MVPA. For child PA at any intensity, parent PA support was significant and positively correlated ($r = 0.23$, $p = 0.033$). Similar to child MVPA, no significant correlation was found between parent PA modelling and child PA at any intensity ($r = 0.15$, $p = 0.179$). Child age was significant and negatively correlated ($r = -0.26$, $p = 0.017$). No other demographic variables were significantly correlated with child MVPA.

**Table 2.** Associations between child demographics, household type, parent PA support, parent PA modelling, and child PA.

|  | Parent PA Support | Parent PA Modelling | Child MVPA | Child PA at any Intensity |
|---|---|---|---|---|
| Parent PA modelling | 0.20 |  |  |  |
| Child MVPA | 0.28 * | −0.02 |  |  |
| Child PA at any intensity | 0.23 * | 0.15 | 0.72 ** |  |
| Child age | 0.01 | 0.12 | −0.18 | −0.26 * |
| Child gender | 0.10 | 0.22 * | −0.01 | 0.16 |
| Child disability group [a] | 0.03 | 0.20 | −0.14 | −0.09 |
| Household type | −0.15 | −0.17 | −0.10 | −0.03 |

Note. MVPA = moderate-to-vigorous physical activity, PA = physical activity. [a] Spearman's rank correlation coefficient. * $p < 0.05$, ** $p < 0.01$.

### 3.3. Parent PA Support and Parent PA Modelling as Predictors of Child MVPA

Table 3 shows the results from the multiple linear regression model for child MVPA. The overall model accounted for approximately 12% of the variance in child MVPA and significantly predicted MVPA (Adjusted $R^2 = 0.12$, $F_{(7,74)} = 2.57$, $p = 0.020$). Parent PA support significantly predicted child MVPA ($\beta = 0.30$, $p = 0.007$), after controlling for the demographic variables. Parent PA modelling was not a significant predictor in the model ($\beta = -0.06$, $p = 0.609$). The P-P plot displayed no substantial departures from normality, and the VIF values showed no evidence of collinearity.

**Table 3.** Relationship between parent support and parent MVPA on the number of days children and youth with disabilities engaged in at least 60 min of MVPA (*N* = 80).

|  | B (SE) | β | *p*-Value | 95% CI | |
|---|---|---|---|---|---|
|  |  |  |  | LLCI | ULCI |
| Parent Support | 0.25 (0.09) | 0.30 | 0.007 | 0.07 | 0.44 |
| Parent MVPA | −0.001 (0.002) | −0.06 | 0.609 | −0.004 | 0.003 |
| Household Type | −0.503 (0.59) | −0.10 | 0.393 | −1.67 | 0.66 |
| Child Age | −0.768 (0.45) | −0.18 | 0.093 | −1.67 | 0.13 |
| Child Gender | −0.263 (0.58) | −0.05 | 0.650 | −1.42 | 0.89 |
| Disability Group: Physical and Sensory [a] | 0.189 (0.67) | 0.03 | 0.778 | −1.14 | 1.52 |
| Disability Group: Other [a] | −1.815 (0.75) | −0.26 | 0.018 | −3.31 | −0.32 |

Note. LLCI = lower-level confidence interval, ULCI = upper-level confidence interval, MVPA = moderate-to-vigorous physical activity. [a] Reference category = Disability Group: Developmental.

### 3.4. Parent PA Support and Parent PA Modelling as Predictors of Child PA at Any Intensity

Table 4 shows the results from the multiple linear regression model for child PA at any intensity. Similar to child MVPA, the overall model explained 9% of the variance of PA at any intensity and significantly predicted child PA at any intensity (Adjusted $R^2 = 0.09$,

$F_{(7,77)} = 2.23$, $p = 0.04$). After controlling for the demographic variables, neither parent PA support ($\beta = 0.19$, $p = 0.078$) nor parent PA modelling ($\beta = 0.12$, $p = 0.274$) significantly predicted child PA at any level. The P-P plot displayed no substantial departures from normality, and the VIF values showed no evidence of collinearity.

**Table 4.** Relationship between parent support and parent MVPA on the number of days children and youth with disabilities engaged in at least 60 min of physical activity at any intensity ($N = 83$).

| | B (SE) | β | *p*-Value | 95% CI | |
|---|---|---|---|---|---|
| | | | | LLCI | ULCI |
| Parent Support | 0.17 (0.10) | 0.19 | 0.078 | −0.02 | 0.36 |
| Parent MVPA | 0.002 (0.002) | 0.12 | 0.274 | −0.002 | 0.01 |
| Household Type | −0.37 (0.60) | −0.07 | 0.540 | −1.58 | 0.83 |
| Child Age | −1.29 (0.49) | −0.29 | 0.010 | −2.26 | −0.32 |
| Child Gender | 0.69 (0.62) | 0.12 | 0.274 | −0.56 | 1.93 |
| Disability Group: Physical and Sensory [a] | −0.71 (0.69) | −0.11 | 0.305 | −2.09 | 0.66 |
| Disability Group: Other [a] | −0.57 (0.82) | −0.07 | 0.489 | −2.19 | 1.06 |

Note. LLCI = lower-level confidence interval, ULCI = upper-level confidence interval, MVPA = moderate-to-vigorous physical activity. [a] Reference category = Disability Group: Developmental.

## 4. Discussion

Parents of CYD play an important role in supporting their child's PA participation [16,17,21–23]. Our study findings add to the existing research by showing that the relationship between parent PA support and the PA participation of CYD may differ during the COVID-19 pandemic, depending on the intensity of the child's PA. Specifically, parent PA support significantly predicted child MVPA, but not PA at any intensity, in CYD. Conversely, parent PA modelling was not a significant predictor of child MVPA or PA at any intensity.

In our study, parent PA support significantly predicted MVPA (defined as activities where the child is out of breath or sweating) but not overall PA (defined as physical activities performed at mild, moderate, and/or vigorous intensities). These results are partially consistent with previous studies that have demonstrated parent PA support to be a significant predictor of PA in CYD [17,18]. In Brown et al.'s [17] study, parent PA support significantly predicted the number of days children and youth with autism spectrum disorder spent in PA at any intensity. However, Siebert et al. [18] found that parent PA support significantly predicted children and youth's time spent in active play (which included both moderate- and vigorous-intensity activities occurring in and out of school). Children's participation in PA can occur in many settings; however, the PA opportunities of CYD have been limited during the pandemic [13], and parents of CYD have had to take on the responsibility of providing these opportunities, often without external support [28]. Notably, organized sport and PA programs were forced to shut down in Canada during the acute period of the pandemic [32]. Speculatively, with organized sport and PA programs no longer available, CYD may have required more support from parents to participate in MVPA, an intensity of PA that they may typically accrue through sports or other organized games. In contrast, participating in light PA (e.g., going for a leisure walk/wheel) may not have changed throughout the pandemic, meaning that parent PA support may not be as heavily relied upon for CYD. Without access to external support including specialized programs, adapted physical education classes, and trained staff support, parents of CYD have had to take on an even greater supporting role compared to before the pandemic. This additional responsibility may explain why parent PA support impacted the PA of CYD differently in the present study compared to other studies conducted prior to the onset of the pandemic [17,18].

As a result of the public health restrictions put in place to help curb the spread of the COVID-19 virus, including the closure of facilities and sport and active recreation

programming, it is possible that the type of support parents of CYD provided in the context of PA was different than what they would have provided before the pandemic. For example, before the pandemic, parent PA support may have included providing transportation to an adapted or inclusive PA or sport program where their child would have the opportunity to engage in MVPA in a more supervised setting. However, without access to PA programs or facilities during the pandemic, parents of CYD had the responsibility of creating and facilitating PA for their child at home, which could present challenges for some parents of CYD. In fact, during the pandemic, parents of CYD have highlighted challenges with supporting their child's PA due to a lack of information and external support available [28]. While PA at any intensity is associated with health benefits for CYD [33], it is recommended that children and youth engage in at least 60 min each day of MVPA for optimal health [8]. Thus, supporting MVPA in addition to PA in general is important for promoting health among children and youth. Our results show parent PA support is positively related to child MVPA. Moving forward, it will be pertinent to provide parents of CYD with the knowledge and skills to support their child's MVPA. By increasing MVPA among CYD, it may be possible to mitigate the physical and mental health challenges faced by CYD as families continue to navigate through the pandemic.

In this study, parent PA modelling was not shown to be a significant predictor of child PA at any intensity or MVPA specifically. These results differ from previous research that has shown parent PA modelling predicts PA in CYD [18,26]. Parents in the current study were about as active (37% meeting MVPA guidelines of 150 min of MVPA per week) as parents of CYD pre-COVID-19 (e.g., 44% meeting guidelines demonstrated by Yazdani et al. [26]). Interestingly, while parents in this study were, on average, engaging in 147 min of MVPA per week (which is close to the MVPA recommendations of at least 150 min per week for Canadian adults), no significant relationship was shown between parent PA modelling and child PA (at any intensity or MVPA specifically). This finding suggests that parent PA modelling may play less of a role in the PA of CYD compared to other factors such as having access to accessible and inclusive spaces to be active [34]. At the very least, our findings suggest that parents of CYD may still be able to influence their child's PA regardless of their own PA levels. Thus, to enhance PA among CYD, efforts should be made to determine strategies that focus on increasing parent PA support behaviours and not necessarily increasing PA among parents of CYD. This may include, but is not limited to, enhancing awareness around the importance of PA at various intensities (including light-, moderate-, and vigorous-intensity activities), providing tangible strategies for parents to support MVPA at home, and ensuring CYD are considered within 'return-to-play' initiatives. Public health and government efforts should be made to ensure that as families of CYD continue to navigate through the pandemic, they are provided with the necessary knowledge, skills, and resources to support the PA of CYD, regardless of the availability of external support (e.g., adapted PA programs and facilities).

While many COVID-19-related restrictions are being lifted across Canada and globally, there remains a considerable lack of sport and recreation programs available to children and youth with and without disabilities [35]. In addition, there may be ongoing health concerns related to in-person programming causing a slower return to play, especially among CYD [13]. Consequently, parents will continue to play an integral role in promoting PA for CYD as Canada works to build back healthy communities. To mitigate the negative impact of COVID-19 on the health of CYD, it will be critical to ensure CYD have access to safe PA opportunities (e.g., PA in outdoor spaces, PA opportunities that allow for physical distancing). These opportunities should extend beyond organized sport and active recreation programs to include opportunities within the home environment. This will be especially important for families who either do not have access to inclusive programs and/or are not yet comfortable to participate in in-person programs due to the ongoing COVID-19-related health risks.

*Limitations and Future Directions*

This study provides novel findings on the relationship between parent PA support behaviours and the PA of CYD during the COVID-19 pandemic. However, there are some limitations that should be considered when interpreting the results. First, the majority of parents had children and youth with developmental disabilities, which prevented us from being statistically powered to analyze the relationships between parent PA support and children and youth's PA across different disability groups (e.g., physical, sensory). Parent PA support behaviours likely differ depending on the type and severity of a child's disability, and thus the results from this study may have differed based on the child's disability group and severity. Future research should focus on disabilities beyond developmental disabilities to better understand how impairment type and severity are related to parent PA support and child PA.

Second, while the survey items included in this study were previously deemed reliable [10], the survey responses may have been impacted by social desirability or recall biases which should be considered when interpreting the results. Finally, the cross-sectional design of the study prevents any cause-and-effect interpretations of the findings. Future longitudinal research is needed to better understand the long-term effects of parent PA support and parent PA modelling on the PA of CYD post-pandemic. The results from this study highlight the need for future research that can develop and test parent PA support interventions in an effort to increase PA among CYD.

**5. Conclusions**

This study demonstrates the important role that parents play in supporting the PA of CYD during the acute period (November 2020) of the COVID-19 pandemic. Parent PA support behaviours were significantly and positively related to the MVPA of CYD. Additionally, our findings suggest that parents themselves do not necessarily have to be physically active to support the PA of CYD. External PA support for CYD remains limited, and thus parents will continue to take on additional responsibilities with regard to supporting PA for their children. Moving forward, practitioners (e.g., occupational, physical, and recreation therapists) should involve parents in programming and treatment and provide tangible strategies for engaging in PA with their child. Interventions targeting parent PA support may provide an opportunity to help mitigate the impact of the pandemic on the PA of CYD as Canada seeks to build back healthy communities post-pandemic. Therefore, future research should focus on developing evidence-based strategies to assist parents with supporting their child's PA.

**Author Contributions:** Conceptualization, K.P.A.-N., S.A.M. and K.A.M.G.; methodology, K.P.A.-N., S.A.M., R.L.B.-G. and K.A.M.G.; formal analysis, M.E.J. and K.P.A.-N.; writing—original draft preparation, M.E.J., N.O., S.A.M. and K.P.A.-N.; writing—review and editing, M.E.J., N.O., S.A.M., R.L.B.-G., K.A.M.G. and K.P.A.-N.; supervision, K.P.A.-N.; project administration, K.P.A.-N.; funding acquisition, K.P.A.-N. and K.A.M.G. All authors have read and agreed to the published version of the manuscript.

**Funding:** This work was supported by a Partnership Grant from the Social Sciences and Humanities Research Council of Canada (Grant no. 895-2013-1021) for the Canadian Disability Participation Project (http://www.cdpp.com/, accessed on 27 July 2022), the Canadian Tire Jumpstart Charities, and a faculty institutional grant from the Faculty of Kinesiology and Physical Education at the University of Toronto and Dalhousie University.

**Institutional Review Board Statement:** The study was conducted according to the guidelines of the Declaration of Helsinki and approved by the Institutional Review Board of the University of Toronto (#31862).

**Informed Consent Statement:** Participants consented to participate at the start of the online survey. This study was approved by the University of Toronto's Research Ethics Board (#31862).

**Data Availability Statement:** For more information regarding data availability, please email Kelly P Arbour-Nicitopoulos, kelly.arbour@utoronto.ca.

**Acknowledgments:** We would like to acknowledge the families in this study for their support on this project.

**Conflicts of Interest:** M.J., N.O., S.A.M., R.B.G., K.A.M.G. and K.A.N. have no conflict of interest to declare.

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
