# Peer review of "Parent Support Is Related to Physical Activity among Children and Youth with Disabilities during the COVID-19 Pandemic: Findings from the National Physical Activity Measurement (NPAM) Study"

_disabilities, doi:10.3390/disabilities2030032_

Round 1
Reviewer 1 Report
Thank you for the opportunity to review the manuscript.
Overall, the research is timely; however, I offer the following suggestions.
The authors use passive language throughout the manuscript. Please edit with an active voice, particularly in the Introduction and the Discussion sections.
Within Methods, identify the developmental disabilities in the sample so it is clear to the reader the breadth and intensity represented as it is pertinent to the range of abilities within the children and youth as well as the related demands on parents to support (such as providing adaptive equipment or techniques).
A clear description of the COVID context is essential, as understanding the constraints on normal behaviors and routines are the key to this research. As part of the introduction (page 2, paragraph 2), relate clearly the limitations and demands in Canada that made it difficult, if not impossible, to maintain a typical routine including group size restrictions, school and business closings, maintaining 6 feet distance, masking, handwashing, remote work at home, quarantine and isolation timelines, etc. While this is current knowledge, future audiences must also clearly understand the breadth and depth of constraints.
As part of the Conclusion, describe directions for future research as well as implications for current and future practice. This is broadly addressed on line 126-129 in the Introduction. However, drawing on the data, answer the “so what” question. What do practioners and researchers do with this knowledge of these relationships (or lack thereof)? How do we help parents prepare? Support materials for parents? In home consultation for parents using distance methodology? Monitor health, weight, mental health, coordination, strength, accuracy, control… of CYD? Coaching teams with teachers, PT, OT, SLP, pediatricians… Lines 372 and 373 are vague and I strongly encourage the authors to offer suggestions.
Introduction:
Line 63 (and again on line 102), Describe participants in the Moore and Moore studies. Were they typically developing youth of the same age? SES? Location?
Lines 64-67, The authors state CYD were disproportionally affected. Please offer specific examples of this when compared to children and youth typically developing (I believe this is the reference group but if I am mistaken, please correct).
Line 71, offer examples of potential contributors to PA
Line 88, Is this research among typically developing children?
Line 146, A survey cannot ask (although a researcher can ask). Suggest “parents responded to questions regarding…”
Line 158 suggest removing “used in this study”
Line 160 suggest removing “through this survey”
Line 310 was this research with typically developing children?
Line 338 Again, please elucidate the specific accommodations that made PA challenging during COVID (for example…)
I am curious to know if parents had the appropriate adaptive equipment at home for their child/youth allowing them to engage in PA. It is hard to know without knowing the range of abilities present in the sample.
Author Response
Parent support is related to physical activity among children and youth with disabilities during the COVID-19 pandemic: Findings from the National Physical Activity Measurement (NPAM) study
Thank you for the opportunity to revise and resubmit our manuscript. There were many great queries raised by the reviewers that we believe have helped the readability of our paper. Below we provide a point-by-point response to each comment:
Reviewer 1
Thank you for the opportunity to review the manuscript.
Overall, the research is timely; however, I offer the following suggestions.
COMMENT: The authors use passive language throughout the manuscript. Please edit with an active voice, particularly in the Introduction and the Discussion sections.
RESPONSE: Thank you for this suggestion to strengthen the readability of our manuscript. We have changed to active voice throughout.
COMMENT: Within Methods, identify the developmental disabilities in the sample so it is clear to the reader the breadth and intensity represented as it is pertinent to the range of abilities within the children and youth as well as the related demands on parents to support (such as providing adaptive equipment or techniques).
RESPONSE: Thank you for raising this important point. We agree that type of disability (beyond the broad categorization such as developmental or physical) and level of severity are important factors to consider. This is especially true when considering caregiver support. In our larger National Physical Activity Measurement study, we collect this information. However, in an effort to reduce participant burden during already stressful times, we removed survey items pertaining to disability severity. In this survey, we only asked parent to specify whether their child’s disability was developmental, physical, sensory or other (if parent selected other they were promoted to specify). To help clarify, we changed “disability type” to “disability group” throughout the paper. In the methods section we state “Parents indicated their child’s disability group as: (1) developmental, (2) physical and/or sensory, or (3) other”.
COMMENT: A clear description of the COVID context is essential, as understanding the constraints on normal behaviors and routines are the key to this research. As part of the introduction (page 2, paragraph 2), relate clearly the limitations and demands in Canada that made it difficult, if not impossible, to maintain a typical routine including group size restrictions, school and business closings, maintaining 6 feet distance, masking, handwashing, remote work at home, quarantine and isolation timelines, etc. While this is current knowledge, future audiences must also clearly understand the breadth and depth of constraints.
RESPONSE: Thank you for bringing this up. We agree that it is important to clearly highlight the extent of the restrictions to accurately contextualize the difficulties families faced with upholding regular routines. We added a few sentences to the beginning of this paragraph to capture the restrictions put in place that impacted physical activity. See lines 62-65 and below for the added information.
“The COVID-19 pandemic introduced several public health restrictions including facility closures (e.g., schools, recreation centers, physiotherapy clinics), limiting size group gatherings (e.g., children’s sport programs) and work-from-home mandates that made it difficult for families to uphold regular daily routines.”
COMMENT: As part of the Conclusion, describe directions for future research as well as implications for current and future practice. This is broadly addressed on line 126-129 in the Introduction. However, drawing on the data, answer the “so what” question. What do practioners and researchers do with this knowledge of these relationships (or lack thereof)? How do we help parents prepare? Support materials for parents? In home consultation for parents using distance methodology? Monitor health, weight, mental health, coordination, strength, accuracy, control… of CYD? Coaching teams with teachers, PT, OT, SLP, pediatricians… Lines 372 and 373 are vague and I strongly encourage the authors to offer suggestions.
RESPONSE: Thank you for providing this feedback and valuable suggestions. We included a few different suggestions for practitioners regarding how they can use this information to better support families. For example, in line 340-347 we discuss the need for enhancing parent awareness for the different intensities of PA, developing appropriate strategies for supporting PA at home and ensuring children and youth with disabilities are included in return-to-play initiatives moving forward to help increase overall PA. We also emphasize the need for safe PA programming for children and youth with disabilities who will remain a vulnerable population. We suggest these programs also move beyond the traditional recreation/sport environment to include home-based PA for children and youth who may not be ready or able to return to group settings (lines 355-360). As per your suggestion, we added these points to the conclusion to further emphasize future directions for both practitioners and researchers. See lines 376-382 and below.
“Moving forward, practitioners (e.g., occupational, physical, and recreation therapists) should involve parents in programming and treatment and provide tangible strategies for engaging in PA with their child. Interventions targeting parent PA support may provide an opportunity to help mitigate the impact of the pandemic on the PA of CYD as Canada seeks to build back healthy communities post-pandemic. Therefore, future re-search should focus on developing evidence-based strategies to assist parents with supporting their child’s PA.”
COMMENT: Line 63 (and again on line 102), Describe participants in the Moore and Moore studies. Were they typically developing youth of the same age? SES? Location?
RESPONSE: The Moore and Moore studies were conducted among children and youth of typical development from across Canada ages 5 to 17 years, and were similar in terms of SES to our cohort. We added this information to the manuscript. Thank you for this suggestion.
COMMENT: Lines 64-67, The authors state CYD were disproportionally affected. Please offer specific examples of this when compared to children and youth typically developing (I believe this is the reference group but if I am mistaken, please correct).
RESPONSE: We have added a few examples from the references paper Armitage & Nellums, 2020: “e.g., reduced service prevision, lack of inclusive health resources”
COMMENT: Line 71, offer examples of potential contributors to PA
RESPONSE: We have added an example of a potential contributer to PA here: “(e.g., the child’s previous engagement in and enjoyment of PA)”
COMMENT: Line 88, Is this research among typically developing children?
RESPONSE: Yes, this line is specifically referring to typically developing children. We have added this here. Thank you for raising this point.
COMMENT: Line 146, A survey cannot ask (although a researcher can ask). Suggest “parents responded to questions regarding…”
RESPONSE: Thank you for bringing this to our attention. We have changed “asked” to “included”. The line now reads:
“This secondary study consisted of the online administration of the 2020 COVID-19 and Childhood Movement Behaviours Survey (developed by Moore et al., 2020) which included questions regarding family demographics, current and previous movement behaviours, and the impact of COVID-19 related restrictions on movement behaviours and health of families.”
COMMENT: Line 158 suggest removing “used in this study” and Line 160 suggest removing “through this survey”
RESPONSE: Thank you for this suggestion to enhance the readability and flow of the paper. We have made these changes in the manuscript.
COMMENT: Line 310 was this research with typically developing children?
RESPONSE: The studies referred to here are specific to CYD which is stated in this sentence: These results differ from previous research that has shown parent PA modelling predicts PA in CYD (Siebert et al., 2017; Yazdani et al., 2013). Thank you for clarifying.
COMMENT: Line 338 Again, please elucidate the specific accommodations that made PA challenging during COVID (for example…)
RESPONSE: Thank you for this suggestion. We have added some examples for what safe PA may look like including outdoor PA opportunities and PA that allows for physical distancing. See lines 344-345.
COMMENT: I am curious to know if parents had the appropriate adaptive equipment at home for their child/youth allowing them to engage in PA. It is hard to know without knowing the range of abilities present in the sample.
RESPONSE: Thank you for raising this important query. While we did not ask these specific questions in the survey, we did conduct follow up interviews with parents of CYD throughout the pandemic. Parents did indicate that it was challenging to participate in physical activity and physiotherapy due to the lack of specialized equipment in the home. The qualitative results from this study can be found in Arbour-Nicitopoulos, K. P., James, M. E., Moore, S. A., Sharma, R., & Martin Ginis, K. A. (2022). Movement behaviours and health of children and youth with disabilities: Impact of the 2020 COVID-19 pandemic. Paediatrics & Child Health, 27(Supplement_1), S66-S71.
Reviewer 2 Report
The paper is succinct, well-written, and clearly articulates the implications of findings. The authors are to be commended on not over-extending the results. Given the unequal sample strata, the authors could delimit the sample to strictly CY with developmental disabilities but I do not see an issue with retaining the full sample if desired. The measurement instrument may partially explain the limited explanatory power of IVs but the finding the parental support is more important to CYD activity levels than parent modeling is clear and useful.
A few minor recommendations for this particular manuscript:
*Line 78 - "parents1" reflects self-citations (copying from a different manuscript)
*Line 193 - the authors made a good choice with their control variables, enhancing the confidence that parental support is more important than parental modeling for this population during COVID-19
*Line 316 - typo regarding 15 minutes of MVPA per week
Author Response
Parent support is related to physical activity among children and youth with disabilities during the COVID-19 pandemic: Findings from the National Physical Activity Measurement (NPAM) study
Thank you for the opportunity to revise and resubmit our manuscript. There were many great queries raised by the reviewers that we believe have helped the readability of our paper. Below we provide a point-by-point response to each comment:
Reviewer 2
The paper is succinct, well-written, and clearly articulates the implications of findings. The authors are to be commended on not over-extending the results. Given the unequal sample strata, the authors could delimit the sample to strictly CY with developmental disabilities but I do not see an issue with retaining the full sample if desired. The measurement instrument may partially explain the limited explanatory power of IVs but the finding the parental support is more important to CYD activity levels than parent modeling is clear and useful.
A few minor recommendations for this particular manuscript:
COMMENT: Line 78 - "parents1" reflects self-citations (copying from a different manuscript)
RESPONSE: Thank you for bringing this to our attention. This is actually indicating a footnote that can be found at the bottom of page 2 which denotes how we define parents in this study.
COMMENT: Line 193 - the authors made a good choice with their control variables, enhancing the confidence that parental support is more important than parental modeling for this population during COVID-19
RESPONSE: Thank you, we appreciate this feedback.
COMMENT: Line 316 - typo regarding 15 minutes of MVPA per week
RESPONSE: Thank you for flagging this. This should have read “150 minutes”. We have made this change in the manuscript on line 316.
Round 2
Reviewer 1 Report
Thank you for your responses and edits to the suggestions. Passive language remains throughout the manuscript; however, the editors will have final authority on style.